# Association between non-acute Traumatic Injury (TI) and Heart Rate Variability (HRV) in adults: A systematic review and meta-analysis

Rabeea Maqsood[1]*, Ahmed Khattab[1], Alexander N. Bennett[2,3], Christopher J. Boos[1,4]

**1** Faculty of Health and Social Sciences, Bournemouth University, Bournemouth, United Kingdom, **2** Academic Department of Military Rehabilitation, Defence Medical Rehabilitation Centre, Loughborough, United Kingdom, **3** National Heart and Lung Institute, Faculty of Medicine, Imperial College London, London, United Kingdom, **4** Department of Cardiology, University Hospital Dorset, NHS Trust, Poole, United Kingdom

* rmaqsood@bournemouth.ac.uk

**Data Availability Statement:** The data collection form and supporting material are available and published as a supporting information.

## Abstract

Heart rate variability (HRV) is a non-invasive measure of autonomic function. The relationship between unselected long-term traumatic injury (TI) and HRV has not been investigated. This systematic review examines the impact of non-acute TI (>7 days post-injury) on standard HRV indices in adults. Four electronic databases (CINAHL, Medline, Scopus, and Web of Science) were searched. The quality of studies, risk of bias (RoB), and quality of evidence (QoE) were assessed using Axis, RoBANS and GRADE, respectively. Using the random-effects model, mean difference (MD) for root mean square of successive differences (RMSSD) and standard deviation of NN-intervals (SDNN), and standardized mean difference (SMD) for Low-frequency (LF): High-Frequency (HF) were pooled in RevMan guided by the heterogeneity score ($I^2$). 2152 records were screened followed by full-text retrieval of 72 studies. 31 studies were assessed on the inclusion and exclusion criteria. Only four studies met the inclusion criteria. Three studies demonstrated a high RoB (mean RoBANS score 14.5±3.31) with a low QoE. TI was associated with a significantly higher resting heart rate. Meta-analysis of three cross-sectional studies demonstrated a statistically significant reduction in RMSSD (MD -8.45ms, 95%CI-12.78, -4.12, p<0.0001) and SDNN (MD -9.93ms, 95% CI-14.82, -5.03, p<0.0001) (low QoE) in participants with TI relative to the uninjured control. The pooled analysis of four studies showed a higher LF: HF ratio among injured versus uninjured (SMD 0.20, 95%CI 0.01–0.39, p<0.04) (very low QoE). Albeit low QoE, non-acute TI is associated with attenuated HRV indicating autonomic imbalance. The findings might explain greater cardiovascular risk following TI.

**Trial registration** PROSPERO registration number: CRD: CRD42021298530.

## Introduction

Traumatic Injury (TI) is known to decrease the quality of life by adversely affecting emotional, physical, and psychological health [1, 2]. It appears to influence autonomic nervous system

**Funding:** This project is a part of RM's Ph.D. studentship- jointly funded by Bournemouth University and the ADVANCE charity, UK. The funders had no role in study design, data collection and analysis, decision to publish, or preparation of the manuscript. There was no additional external funding received for this study.

**Competing interests:** The authors have declared that no competing interests exist.

(ANS) responses by enhancing sympathetic tone and attenuating the parasympathetic activity in the body [3]. This can be clinically investigated using the variation in time intervals between consecutive heartbeats—heart rate variability (HRV) [4]. HRV is an established objective, non-invasive and indirect measure of autonomic function [5, 6]. In acute trauma settings, HRV has emerged as a new vital sign and triage tool [7–9]. In addition to being a robust cardiovascular marker to predict major adverse cardiovascular events and all-cause mortality [10], lower HRV is inversely linked to injury severity across a wide variety of injury types [11–13].

The adverse physiological and psychological health effects of TI have been reported in civilian [14, 15] and military populations [16]. In the last two decades, research into the impact of TI on HRV has been largely focused on acute trauma and during hospital admission [17–20]. Examination of the relationship between non-acute T1 and HRV has predominantly been in selected injured populations with the spinal cord, head/brain, and psychological trauma (post-traumatic stress disorder-PTSD and depression) [21–24]. While systematic reviews have been conducted previously to assess the association between HRV and traumatic brain injury [25], spinal cord injury [26], PTSD [27] and depression [28], the longer-term effects of unselected TI on HRV have not been reviewed and warrant further examination [29]. With a recent study reporting an increase in cardiovascular disease (CVD) risk in injured military veterans [30], understanding the health outcomes among injured military and civilian populations following a traumatic injury is essential.

This systematic review aims to answer our research question using Population, Exposure, Outcome (PECO) framework [31]: What is the association between non-acute TI *(E)* and HRV *(O)* in adults with unselected TI *(P)* versus uninjured controls *(C)*?

## Methods

The protocol of this systematic review is registered at PROSPERO (CRD: CRD42021298530. The protocol for this systematic review has been recently published [32] (S1 Protocol). This systematic review was conducted using the Preferred Reporting Items for Systematic reviews and Meta-Analyses (PRISMA) 2020 guidelines and checklist [33] (S1 Table).

### Eligibility criteria

**Population.** We included published studies of any design which had adult human participants (aged 18 or above) with sustained TI >7 days previously. The rationale was to include a broad spectrum of participants (civilian and military) [14–16].

**Exposure.** The selection of TI >7 days as the cut-off point was informed by a previous review on mild traumatic brain injury [34] and to mitigate the impact of the early physiological impact of acute trauma on HRV [17–20]. Studies with participants of traumatic brain injury, spinal cord, depression, and post-traumatic stress disorder were excluded [26, 35–37]. Similarly, studies that reported acute injuries (which led to death upon hospital admission), and with adolescents, animals, or children as participants were excluded [17–20].

**Comparator.** Only studies with a comparator or a control group were eligible for inclusion.

**Outcome.** As a primary outcome of this review, only recognised and established measures of HRV were eligible for inclusion [6] (S2 Table). The primary outcomes were limited to the most widely reported time and frequency domain measures of HRV [6]. Our secondary outcomes were resting heart rate (HR) among included HRV studies, type of injury, and time from injury.

## Information sources

A systematic literature search was undertaken by two reviewers (RM and CJB) in 4 electronic bibliographic databases: CINAHL, Medline, Scopus, and Web of Science, in addition to the Cochrane library. The Medical Subject Headings (MeSH) and CINAHL headings were used in Medline and CINAHL, respectively for trauma, wounds, injury, and heart rate variability. The title and abstract fields were searched with a limit on language (only English). There were no restrictions on date and geographical area. The last search was conducted on 10 August 2022.

## Search strategy

The master search strategy was developed by two reviewers (RM and CJB) and was adapted according to each database, with the help of an experienced librarian. Following is the search strategy for Web of Science:

trauma* OR wound* OR "blast" OR explosion* OR injur* AND "heart rate variability" OR "HRV" OR "heart rate variation*" OR "heart rate complexity" OR "SDNN" OR "RMSSD" OR "autonomic function*" OR "autonomic reactivity" OR "HR-variability" OR "autonomic regulation" OR "autonomic activity".

## Selection process

The titles and abstracts were screened by two reviewers (RM and CJB) independently. The reference lists of included studies were also hand-searched to supplement the searches and include studies that might have been missed otherwise. Disagreements were resolved by discussion between two reviewers (RM and CJB), inviting the third reviewer (AK) wherever needed. Mendeley (version 1.19.8) was used to manage records and de-duplications of the results. The full texts were read and evaluated by the reviewers (RM and CJB), independently. Supplementary probes were used at the initial stage of the screening (S3 Table). In case of a disagreement at any stage, a third reviewer (AK) was invited to reach a consensus.

## Data collection process

The data-extraction form (S4 Table) was developed in Microsoft Excel following the guidelines set by the Centre for Reviews and Dissemination [38]. The form was continually adapted during the initial data-extraction stage to extract the relevant data at best and was also tested a priori to the final data-extraction stage to ensure its reliability. The lead reviewer (RM) extracted the data items using the data-extraction form. The second reviewer (CJB) cross-checked the data extraction. Both reviewers (RM and CJB) discussed the final data extraction. All records were maintained in Mendeley (version 1.19.8)

## Data items

The following data (not limited to) were extracted from each study: date of publication, authors, title, setting, country, study aims, and design, sample size, participants' sex, and age, exposure and outcome variables, type of traumatic injury, injury-severity measure, time from injury, resting HR and HRV. The most commonly reported HRV measures were included in the meta-analysis as a primary outcome. Heart rate was reported as a secondary outcome in the narrative synthesis. The corresponding author of one study was contacted to access the missing data.

## Study risk of bias assessment

Two reviewers (RM and CJB) performed the critical appraisal of the studies using Axis- a critical appraisal tool developed for cross-sectional studies [39]. Studies scoring <10, 10–15, >15

were considered low, moderate, and high-quality studies, respectively as previously used [40]. The Risk of Bias (RoB) of the included studies was assessed using the Risk of Bias Assessment of Non-randomized Studies (RoBANS) [41] by two reviewers (RM and CJB), independently. Based on the criteria used in a previous study [40], studies scoring 0, 0–2, >2 were rated as having a low, moderate, and high risk of bias. The disagreements were resolved by a discussion with the third reviewer (AK).

### Effect measures

Mean difference (MD) was used as a summary statistic for continuous variables (HRV measures), with a 95% Confidence Interval (CI). The means and standard deviations (SD) reported for the exposure and outcome measures were extracted. The means and SD of two or more two groups were calculated using Cochrane's recommended method [42].

### Synthesis methods

All statistical analyses were performed in Review Manager (RevMen 5.4) [43]. Where permitted by data, outcome measures of included studies were pooled to perform a meta-analysis using random effects [44] -which is based on the a priori assumption of heterogeneity in HRV measurement and recording time across studies. The level of significance was set at $p < 0.05$. Using narrative synthesis, the characteristics of included studies were tabulated. Sub-group analysis was not possible due to the limited number of included studies.

### Certainty assessment

The overall quality of evidence (QoE) was assessed using Grading of Recommendations Assessment, Development, and Evaluation (GRADE) with GRADEproGDT [45]. GRADE is a ranking system to assess the quality of evidence included in a systematic review [46]. Scoring was upgraded by one level if 50% of studies had a large effect size, dose-response gradient, and minimised confounding effect [46].

## Results

### Study selection

In total, 3839 records were identified with the only limit being on language (English). After the removal of duplicates (n = 1687), a further 2080 records were excluded based on their irrelevancy to eligible study design, publication status, age group, and exposure. 72 studies were retrieved for full text and evaluated against the inclusion and exclusion criteria. Only four studies met the inclusion criteria and were included in the quantitative meta-analysis. Fig 1 shows the PRISMA Flow Diagram [33].

### Study characteristics

The four studies were included in this review and together represented 258 participants with a non-acute TI (exposed group) and 203 uninjured participants (control group). One study was longitudinal [47], one was a case-control [48] and two were cross-sectional [49, 50]. The types of injury reported in the studies were traffic accidents [47], Whiplash Associated Disorder (WAD) (neck injury) [48], traumatic amputation [49], and burns [50]. The mean age range of participants with and without injury was 43.6–56.4 years and 41.7–59.4 years, respectively. Time from injury varied across studies from six weeks [47] to more than twenty years [49]. Overall, the majority of participants were men and middle-aged except for one study with

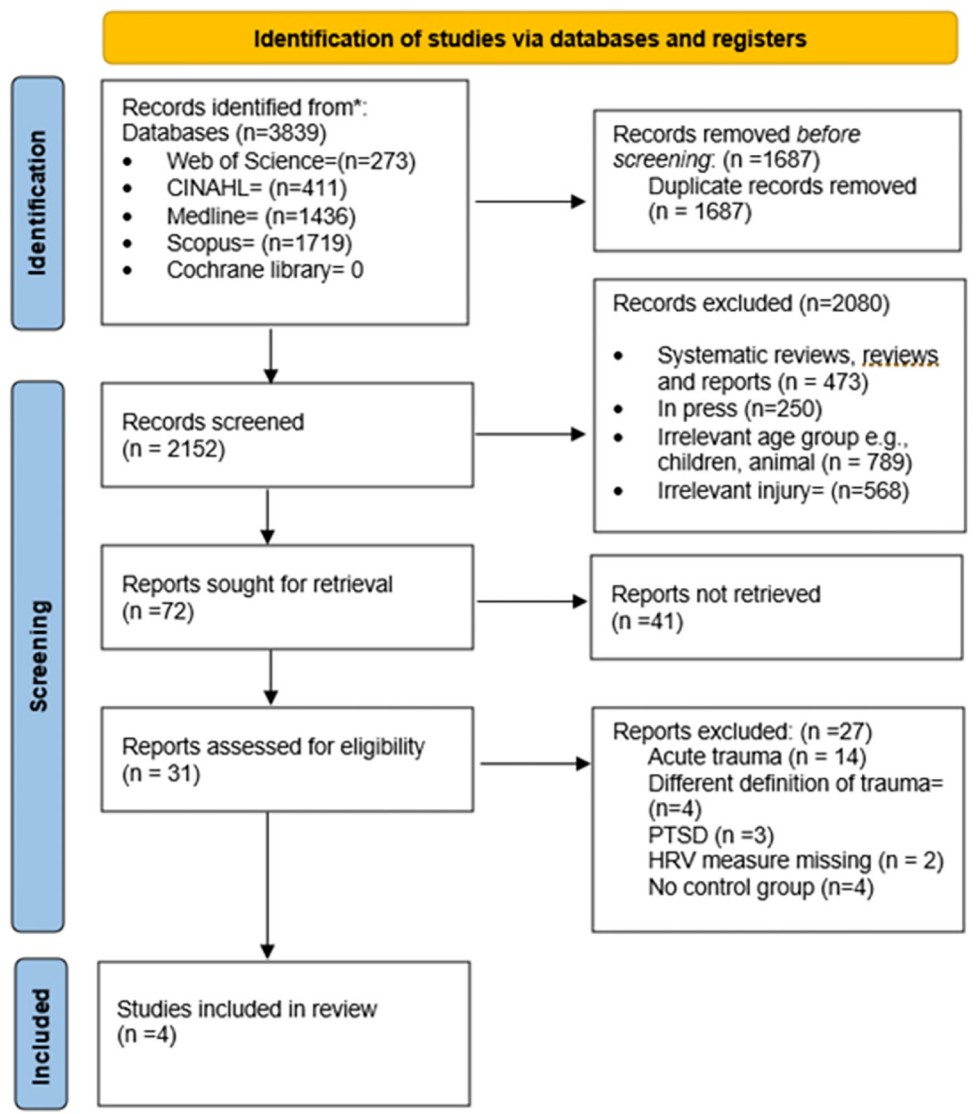

**Fig 1. The PRISMA flow diagram.**

older participants (age >50 years) [49]. One study had military servicemen in the exposed group [49] as opposed to civilians of both sexes in the other three studies (Table 1).

Three studies collected HRV data using electrocardiography (ECG) as opposed to one study with Photoplethysmography (PPG) [48]. Two studies reported the posture maintained during the HRV measurement [47, 49]. In the three ECG-acquired HRV studies, the duration of recording was five minutes in one [49], 10 minutes in another [47], and 24 hours in the third study [50]. Significant variations in posture during HRV measurement were observed in all studies. Two studies reported posture with participants in the sitting position [47] and supine and standing [49]. Two studies did not report the posture while one study also had participants perform their daily activity during a 24-hour recording of HRV [50].

The most frequently reported HRV measures in all four studies were High Frequency (HF), Low Frequency (LF) power followed by the standard deviation of NN intervals (SDNN), the root mean square of successive differences between normal heartbeats (RMSSD), and LF: HF ratio (Table 2). Resting HR was reported in all 4 studies.

Table 1. The characteristics of studies included in the systematic review.

| Author(s) | Location | Study Design and sample size | Sample (n) | Participant's mean age (years) | Exposure (Assessment unit*) | Time from injury | Outcome Measured | Recording type and time | Primary Outcome (HRV) | Secondary outcome (HR) |
|---|---|---|---|---|---|---|---|---|---|---|
| Peles et al. 1995 | Israel | Cross-sectional, observational (n = 105) | E = Servicemen in Israel Defence forces who had traumatic lower limb amputation (n = 52) C = civilians recruited from another ongoing study (n = 53) Male | E = 56.4 ±5.2 C = 59.5 ±4.8 | Traumatic lower limb amputation Not reported* | Wounded between 1948 and 1974 during service | Autonomic function (HRV) and insulin resistance in both groups | ECG- 5 min At: Baseline (supine), orthostatic, and after glucose administration | Higher LF and HF in amputees than in the controls Slightly higher LF: HF ratio in amputees than the controls | Slightly higher HR in amputees than in controls |
| De Kooning et al. 2013 | Belgium | Case-control (n = 61) | E = patients with chronic WAD(n = 30) C = Healthy people without WAD (n = 31) Male and Female | E = 43.6 ± 9.44 C = 43.45 ±15.87 | WAD Quebec Task force classification * | Chronic symptoms of WAD at least for 3 months | Autonomic response (HRV and skin conductance) in WAD patients in response to pain | Photoplethysmography (PPG) Inter-beat interval of 400ms -1400 ms | Lower baseline SDNN, RMSSD, LF, HF in WAD patients than in healthy controls Higher LF: HF ratio in WAD patients than in healthy controls | Higher HR in WAD patients than in healthy controls |
| Joo et al. 2018 | Korea | Cross-sectional, observational (n = 70) | E = Patients with electrical, major, and minor burns (n = 60) C = Healthy controls without burns Male and female | E = 45.70 ±11.65 C = 41.7 ±2.2 | Burn injury (Electrical burns, major and minor burns) %total body surface area* | 81.03 ±26.57 days | Autonomic function (HRV) Time and Frequency domain | ECG, 24-hour | Lower LF, HF, SDNN, RMSSD in burn patients than healthy controls | Higher HR in people with burns than healthy controls |
| Pozzato et al. 2021 | Australia | Longitudinal (3-6-12 weeks post-injury follow-up) N = 232 | E = Injured adults (N = 120) C = Uninjured healthy controls (n = 112) Male and female | E = 45.17 ±16.99 C = 45 ±17.5 | Traffic-related traumatic injury ISS* | Sustained in the last 6 weeks | HRV and skin conductance | ECG- 5 min | Lower HRV (all parameters) in injured than in uninjured healthy controls | Higher HR in the injured group than the uninjured group. |

Abbreviations: Heart Rate Variability (HRV), Whiplash Associated Disorder (WAD), Injury Severity Score (ISS), Electrocardiography (ECG), Heart Rate (HR), Low Frequency (LF), High Frequency (HF), standard deviation of NN intervals (SDNN), the root mean square of successive differences between normal heartbeats (RMSSD)

**Table 2. The most reported heart rate variability parameters in the included studies and the direction of effect.**

| Study | Time Domain Measures | | Frequency Domain Measures | | |
|---|---|---|---|---|---|
| | SDNN | RMSSD | HF | LF | LF/HF |
| Peles et al. 1995 | - | - | ↑ | ↑ | Slightly ↑ |
| De Kooning et al. 2013 | ↓ | ↓ | ↓ | ↓ | ↑ |
| Joo et al. 2018 | ↓ | ↓ | ↓ | ↓ | ↑ |
| Pozzato et al. 2021 | ↓ | ↓ | ↓ | ↓ | Slightly ↓ |

Note: Direction of means represented as (↑ high, ↓ low) for the exposed group compared with the mean of the control group from each study.

## Risk of bias in studies

The scores from the Axis critical appraisal tool for each study are reported (S5 Table). Only one study was of high quality [47] and three were of moderate quality. The quality scores ranged from 11–19 (out of a maximum of 20). The mean quality score was 14.5 ± 3.31. Regarding the risk of bias using RoBANS, only one study had a low risk of bias [47] (S6 Table).

## Results of syntheses

**Heart Rate Variability.** RMSSD, SDNN, and LF: HF ratio were chosen for meta-analysis because of the homogeneity of their units reported in most studies and their wide use in the field [6]. Since in all three studies, RMSSD and SDNN were reported in the same unit, mean difference (MD) was used to pool the data as also suggested [51], whereas standardised mean difference (SMD) was used to pool data for LF: HF ratio, guided by the heterogeneity score [44].

Heterogeneity was insignificant ($I^2 = 0$) for all outcomes. The meta-analysis of three studies indicated a large and statistically significant difference in time domain indices of HRV between injured and uninjured groups. RMSSD was significantly lower in injured than uninjured groups (MD -8.45 ms, 95%CI -12.78, -4.12, p<0.0001) (Fig 2) SDNN was also significantly lower in injured relative to uninjured groups (MD -9.93 ms 95%CI -14.82, -5.03, p<0.0001) (Fig 3). LF: HF was higher in the injured than uninjured group (SMD 0.20 95%CI 0.01–0.39, p<0.05) (Fig 4).

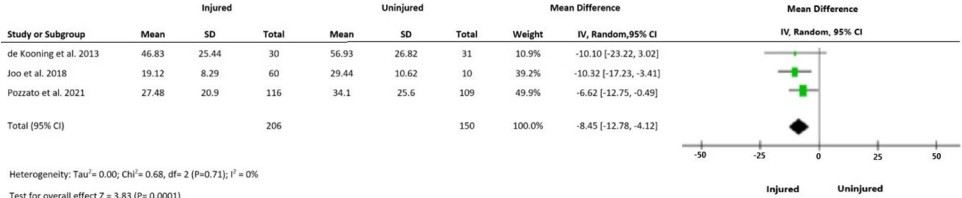

**Fig 2. Outcome 1: Difference in RMSSD (ms) between injured and uninjured groups.**

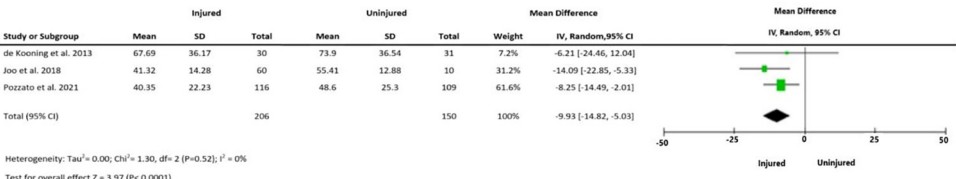

**Fig 3. Outcome 2: Difference in SDNN (ms) between injured and uninjured groups.**

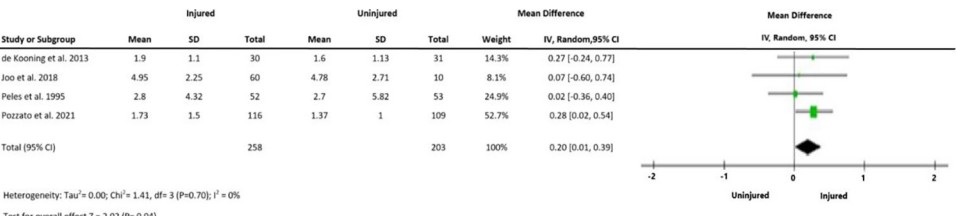

**Fig 4. Outcome 3: Difference in LF: HF ratio between injured and uninjured groups.**

## Reporting biases

Funnel plot analysis revealed no asymmetry, but considerable variation in Standard Error (SE) across studies for all outcomes suggesting a degree of systematic bias (S1 Fig). The publication bias (Egger's test) could not be performed owing to the limited number of studies [52].

## Certainty of evidence

For outcomes SDNN and RMSSD, the QoE assessed was low, whereas for LF: HF ratio it was very low. The GRADE evidence profile and summary of findings tables are given (S7 Table).

## Discussion

To the authors' knowledge, this is the first systematic review to examine the relationship between non-acute generalised T1 and HRV. We identified four suitable studies and observed that a non-acute TI was associated with lower RMSSD and SDNN, and higher LF: HF ratio and resting HR compared with that of the uninjured control group. Overall, the quality of evidence in support of the findings was relatively low with a high risk of bias.

There are several potential mechanisms to explain the decline in HRV following TI. Perhaps the most obvious is that of physical deconditioning following TI and the associated reduction in physical activity associated with TI-related physical injury [53]. Another plausible mechanism is chronic low-grade inflammation which may be persistent leading to tissue degeneration [54] and subsequent autonomic imbalance. The results of the recently reported Armed Services Trauma Rehabilitation Outcome Study (ADVANCE) which examined the relative cardiovascular risk profiles of 579 injured versus 565 matched (age, sex, deployment) uninjured military personnel showed that the levels of high-sensitivity C-reactive protein (HS-CRP) were significantly greater among those with TI [26]. In the ADVANCE study it was also observed that T1 injury was associated with higher resting HR, relative obesity, dyslipidaemia, and insulin resistance among the injured [30]- which are all reported to be inversely related to HRV [55, 56]. Another plausible mechanism to explain the lower HRV with TI may be related to the higher burden of depression, anxiety, and PTSD which are all known to reduce HRV [57]. It remains uncertain as to whether combat-related TI further enhances these negative and inverse relationships and remains the focus of the ongoing ADVANCE study [58].

HRV is predominately controlled by the ANS and acts as an indicator of autonomic activity [4]. The balance between the sympathetic and parasympathetic branches of ANS is essential for homeostasis and overall physiological and psychological well-being [59, 60]. The results of this systematic review would suggest that non-acute TI leads to a relative sympatho-vagal dominance explaining the higher LF: HF ratio among the injured. The lower RMSSD among the injured would also suggest their greater vagal withdrawal. RMSSD has been shown to strongly correlate with HF power and is preferred over HF power as a measure of vagal tone because it is free of respiratory influence [6, 61]. These findings are of major importance given the

recognised link between increased LF: HF ratio and all-cause mortality among patients with traumatic brain injury [25] and haemodynamically stable unselected acute TI [62]. The longer-term association between HRV measured among non-acute TI remains uncertain and is the primary aim of the ongoing ADVANCE Study [58, 63].

This systematic review has several strengths which need to be acknowledged. To date, it is the first systematic review to examine the association between non-acute TI and HRV in adults. This systematic review has been conducted in a transparent manner using the PRISMA guidelines since its registration on the PROSPERO database and the publication of its protocol [32]. The quality of evidence has also been assessed using GRADE. Another strength of this study is its extensive literature search undertaken across four electronic databases.

Nevertheless, the results of this systematic review should be interpreted with caution. All included studies had a high risk of bias as assessed through RoBANS except one [47] and had a methodological quality ranging from low to moderate. Due to the limited number of studies, sensitivity and subgroup analyses could not be performed. For one study, the data points were extracted from the graphical figures as the original paper had not provided the data, and contact with the author was not possible [49]. We also observed methodological heterogeneity in HRV measurement, recording, and analysis across studies. Except for one study, all studies were cross-sectional. Our pooled estimate does not account for changes introduced by different postures in all studies. Similarly, albeit inappropriate, HRV measures were pooled despite differences in HRV measurement durations e.g., 5 min, 10 min, and 24 hours and type of injury (burn, traffic accident, WAD, and amputation). Notably, non-linear indices of HRV were not included and pooled in the meta-analysis owing to the limited availability of data. The quality of evidence assessed through GRADE is low- which means further research is very likely to have an important impact on our confidence in the estimate of effect and is likely to change the estimate. Lastly, the generalisability of the conclusion is limited due to the relatively small sample size and the number of studies included in the meta-analysis.

Nonetheless, this systematic review provides small yet important evidence on the effect of unselected non-acute TIs on HRV. Timely detection of autonomic imbalance is pertinent to informing acute trauma care [3]. The results of this systematic review have shown that chronic TI is associated with a reduction of HRV which is an indirect marker of reduced recovery and increased cardiovascular risk. This may have translational applications for trauma-care practice including targeted interventions for the most vulnerable survivors of TI. For future research, longitudinal studies are warranted to examine the causal relationship between non-acute traumatic injuries and HRV (including both linear and non-linear indices) in civilian and military populations, separately. Within this scope, there is an ongoing 20-year longitudinal study called Armed Services Trauma Rehabilitation Outcome (ADVANCE) which aims to examine the long-term impact of CRTI on CVD risk including HRV monitoring [58, 63]. Having a similar study with civilians may also help better understand the mechanism and factors which affect TI-HRV association in different populations.

## Conclusion

This systematic review and meta-analysis have shown that non-acute unselected TI is associated with lower RMSSD and SDNN, and higher LF: HF ratio and resting heart rate than that of matched uninjured controls. Larger high-quality longitudinal studies are warranted to investigate the causal relationship between non-acute TI and HRV.

## Supporting information

**S1 Protocol. Systematic review protocol.**
(PDF)

**S1 Fig.** Funnel plot for outcomes SDNN (A), RMSSD (B) and LF: HF ratio (C).
(DOCX)

**S1 Table. PRISMA 2020 checklist for systematic review.**
(DOCX)

**S2 Table. The inclusion and exclusion criteria.**
(DOCX)

**S3 Table. The supplementary probs for initial screening.**
(DOCX)

**S4 Table. The data extraction form.**
(DOCX)

**S5 Table. The Axis quality appraisal scores for the included studies.**
(DOCX)

**S6 Table. The Risk of Bias score of each study using Risk of Bias Assessment tool for Non-randomized Studies.**
(DOCX)

**S7 Table. GRADE Evidence Profile (a) and summary of findings table (b) of Outcome Measures SDNN, RMSSD and LF: HF ratio.**
(DOCX)

## Acknowledgments

We would like to acknowledge the support of Professor Paul Cullinan, Professor Nicola Fear, Professor Anthony Bull, Susie Schofield, and Emma Coady.

## Author Contributions

**Conceptualization:** Rabeea Maqsood, Christopher J. Boos.

**Data curation:** Rabeea Maqsood.

**Formal analysis:** Rabeea Maqsood.

**Funding acquisition:** Alexander N. Bennett, Christopher J. Boos.

**Investigation:** Rabeea Maqsood.

**Methodology:** Rabeea Maqsood, Christopher J. Boos.

**Project administration:** Rabeea Maqsood.

**Resources:** Rabeea Maqsood.

**Software:** Rabeea Maqsood.

**Supervision:** Ahmed Khattab, Alexander N. Bennett, Christopher J. Boos.

**Visualization:** Rabeea Maqsood.

**Writing – original draft:** Rabeea Maqsood.

**Writing – review & editing:** Rabeea Maqsood, Ahmed Khattab, Alexander N. Bennett, Christopher J. Boos.

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
