## [Decision Letter · Decision Letter 0]

21 Nov 2022

PONE-D-22-26222Association between non-acute Traumatic Injury (TI) and Heart Rate Variability (HRV) in adults: a systematic review and meta-analysisPLOS ONE

Dear Dr. Maqsood,

Thank you for submitting your manuscript to PLOS ONE. After careful consideration, we feel that it has merit but does not fully meet PLOS ONE’s publication criteria as it currently stands. Therefore, we invite you to submit a revised version of the manuscript that addresses the points raised during the review process. Please submit your revised manuscript by Jan 05 2023 11:59PM. If you will need more time than this to complete your revisions, please reply to this message or contact the journal office at plosone@plos.org. Please include the following items when submitting your revised manuscript:A rebuttal letter that responds to each point raised by the academic editor and reviewer(s). You should upload this letter as a separate file labeled 'Response to Reviewers'.A marked-up copy of your manuscript that highlights changes made to the original version. You should upload this as a separate file labeled 'Revised Manuscript with Track Changes'.An unmarked version of your revised paper without tracked changes. You should upload this as a separate file labeled 'Manuscript'.

We look forward to receiving your revised manuscript.

Kind regards,

Tariq Jamal Siddiqi

Academic Editor

PLOS ONE

Journal Requirements:

"This project is a part of RM’s Ph.D. studentship- jointly funded by Bournemouth University and the ADVANCE charity, UK."

"This project is a part of RM’s Ph.D. studentship- jointly funded by Bournemouth University and the ADVANCE charity, UK."

Reviewers' comments:

Reviewer's Responses to Questions

**Comments to the Author**

1. Is the manuscript technically sound, and do the data support the conclusions?

Reviewer #1: Yes

2. Has the statistical analysis been performed appropriately and rigorously? 

Reviewer #1: Yes

3. Have the authors made all data underlying the findings in their manuscript fully available?

Reviewer #1: Yes

4. Is the manuscript presented in an intelligible fashion and written in standard English?

Reviewer #1: Yes

5. Review Comments to the Author

Reviewer #1: Rabeea et al. conducted a study on "Association between Non-Acute Traumatic Injury (TI) and Heart Rate Variability (HRV) in Adults: A Systematic Review and Meta-analysis,"  in which they investigated the impact of non-acute TI on standard HRV indices in adults. This study can be improved by incorporating the following points:

In the introduction, the authors should highlight how this study addresses a current gap in the literature..

In the introduction, authors can also discuss what current guidelines indicate regarding the use of HRV indices in the management of non-acute injuries caused by trauma.

This sentence needs to be rewritten in the introduction: "What is the relationship between non-acute TI (E) and HRV (O) in people with unselected TI (P) compared to people who haven't been hurt (C)?"

A few typing errors need to be reviewed.

In the limitations section, authors should also discuss that non-linear HRV indices were not included in this study owing to the limited availability of data and that additional research is required to evaluate the effect of non-acute TI on standard HRV indices in civilian and military individuals separately.

6. PLOS authors have the option to publish the peer review history of their article (what does this mean?). If published, this will include your full peer review and any attached files.

Reviewer #1: No

---

## [Author Response · Author response to Decision Letter 0]

30 Nov 2022

The following has also been uploaded as a separate file. 

PONE-D-22-26222

Association between non-acute Traumatic Injury (TI) and Heart Rate Variability (HRV) in adults: a systematic review and meta-analysis

We would like to thank the academic editor and the reviewer for providing insightful feedback on our manuscript. The suggestions made by the academic editor and the reviewer have been incorporated into the manuscript and are highlighted. Please see below, in blue, for a point-by-point response to the reviewer’s comments. We hope that this manuscript might now be considered suitable for publication.

Journal Requirements:

Response: Thank you for bringing this to our attention. I have checked the format and –to the best of our knowledge- the manuscript has now been revised and written according to PLOS One’s style requirements, including those for file naming, authors’ affiliations, and supporting information. The figure captions have also been added in the manuscript (Line 203 & Lines 271-276 and figures have been uploaded in TIFF format, as per PLOS One’s format requirement. 

"This project is a part of RM’s Ph.D. studentship- jointly funded by Bournemouth University and the ADVANCE charity, UK."

Response: Thank you for this comment. The revised funding statement has been stated in the cover letter as recommended by the journal: 

This project is a part of RM’s Ph.D. studentship- jointly funded by Bournemouth University and the ADVANCE charity, UK. The funders had no role in study design, data collection and analysis, decision to publish, or preparation of the manuscript. There was no additional external funding received for this study. 

Response: Noted. The Funding Statement has now been amended with the cover letter.

"This project is a part of RM’s Ph.D. studentship- jointly funded by Bournemouth University and the ADVANCE charity, UK."

Response: Thanks for highlighting this. The revised funding statement has been mentioned in the cover letter. Please see the response to comment 2. 

Response: Thank you, the Role of Funder has now been amended in the cover letter. 

Response: Thank you for this suggestion. However, this is a systematic review of existing evidence. No new dataset was generated. The data (mean differences/ standardised mean difference between injured and control groups) were extracted from the included studies and pooled in RevMan. The studies included in the systematic review have been cited and referenced in the manuscript- the readers can access those studies to access original data if needed.

Please note the data extraction forms and the data resulting from the critical appraisal, risk of bias assessment, and strength of evidence have been uploaded as supporting material and are available for the readers. 

Response: Not applicable. 

Review Comments to the Author:

Reviewer #1: Rabeea et al. conducted a study on "Association between Non-Acute Traumatic Injury (TI) and Heart Rate Variability (HRV) in Adults: A Systematic Review and Meta-analysis," in which they investigated the impact of non-acute TI on standard HRV indices in adults. This study can be improved by incorporating the following points:

In the introduction, the authors should highlight how this study addresses a current gap in the literature..

Response: Thank you for this comment. We would like to draw your attention to the following section in the Introduction which answers this question- how does this study address a current gap in the literature?

Lines 81-88: “Examination of the relationship between non-acute T1 and HRV has predominantly been in selected injured populations with the spinal cord, head/brain, and psychological trauma (post-traumatic stress disorder-PTSD and depression) [21, 22, 23, 24]. While systematic reviews have been conducted previously to assess the association between HRV and traumatic brain injury [25], spinal cord injury [26], PTSD [27] and depression [28], the longer-term effects of unselected TI on HRV have not been reviewed and warrant further examination [29]”.

The above section highlights the lack of a systematic review on unselected TI and HRV- what this systematic review aims to address. 

In the introduction, authors can also discuss what current guidelines indicate regarding the use of HRV indices in the management of non-acute injuries caused by trauma.

Response: Thanks for this suggestion. While we have adhered to the guidelines of the Task Force for reporting HRV indices throughout this systematic review, to the authors’ knowledge, there is no guideline on the use of HRV indices in the management of non-acute TI and trauma. As mentioned in the introduction section, several studies have explored the impact of acute trauma (immediately upon hospital admission) and selected TI (spinal cord/brain injury) on HRV, but no guidelines exist due to limited evidence on long-term HRV and non-acute TIs. This has been highlighted in the ‘future research section’ of our manuscript. Furthermore, at present, the use of HRV in non-acute traumatic injury is restricted to research studies. However, HRV has been widely adopted into decision-making and training protocols among athletes in everyday practice. With further research, we strongly believe that the use of HRV parameters will be integrated into clinical practice and for the rehabilitation of patients with a traumatic injury.

This sentence needs to be rewritten in the introduction: "What is the relationship between non-acute TI (E) and HRV (O) in people with unselected TI (P) compared to people who haven't been hurt (C)?"

Response: We appreciate this suggestion. However, the terms “people” and “hurt” may be too generic in terms of Population, exposure, control, and outcome (PECO) format. We would like to stick with the original sentence (lines 93-95): What is the association between non-acute TI (E) and HRV (O) in adults with unselected TI (P) versus uninjured controls (C)? because it focuses on a particular population (adults) and injury status (injured vs uninjured). 

A few typing errors need to be reviewed.

Response: Thanks for highlighting this. The manuscript has now been proofread for typing/grammatical errors and the changes have been made in both (tracked and untracked) versions of the manuscript. 

In the limitations section, authors should also discuss that non-linear HRV indices were not included in this study owing to the limited availability of data and that additional research is required to evaluate the effect of non-acute TI on standard HRV indices in civilian and military individuals separately.

Response: Thank you for this suggestion. The above sentences have been included in the limitation (lines 343-344) and future research sections (lines 358-359), respectively.

---

## [Decision Letter · Decision Letter 1]

8 Jan 2023

Association between non-acute Traumatic Injury (TI) and Heart Rate Variability (HRV) in adults: a systematic review and meta-analysis

PONE-D-22-26222R1

Dear Dr. Maqsood,

We’re pleased to inform you that your manuscript has been judged scientifically suitable for publication and will be formally accepted for publication once it meets all outstanding technical requirements.

Kind regards,

Tariq Jamal Siddiqi

Academic Editor

PLOS ONE

Additional Editor Comments (optional):

Reviewers' comments:

Reviewer's Responses to Questions

**Comments to the Author**

1. If the authors have adequately addressed your comments raised in a previous round of review and you feel that this manuscript is now acceptable for publication, you may indicate that here to bypass the “Comments to the Author” section, enter your conflict of interest statement in the “Confidential to Editor” section, and submit your "Accept" recommendation.

Reviewer #1: All comments have been addressed

2. Is the manuscript technically sound, and do the data support the conclusions?

Reviewer #1: Yes

3. Has the statistical analysis been performed appropriately and rigorously? 

Reviewer #1: Yes

4. Have the authors made all data underlying the findings in their manuscript fully available?

Reviewer #1: Yes

5. Is the manuscript presented in an intelligible fashion and written in standard English?

Reviewer #1: Yes

6. Review Comments to the Author

Reviewer #1: (No Response)

7. PLOS authors have the option to publish the peer review history of their article (what does this mean?). If published, this will include your full peer review and any attached files.

Reviewer #1: No

---

## [Editor Report · Acceptance letter]

12 Jan 2023

PONE-D-22-26222R1 

Association between non-acute Traumatic Injury (TI) and Heart Rate Variability (HRV) in adults: a systematic review and meta-analysis 

Dear Dr. Maqsood:

I'm pleased to inform you that your manuscript has been deemed suitable for publication in PLOS ONE. Congratulations! Your manuscript is now with our production department. 

Kind regards, 

on behalf of

Dr. Tariq Jamal Siddiqi 

Academic Editor

PLOS ONE